# Investigation on the Effective Measures for Improving the Performance of Calorimetric Microflow Sensor

**DOI:** 10.3390/s23177413

**Published:** 2023-08-25

**Authors:** Jiali Qi, Chun Shao, Wei Wu, Ruijin Wang

**Affiliations:** School of Mechanical Engineering, Hangzhou Dianzi University, Hangzhou 310018, China; qijiali@hdu.edu.cn (J.Q.); weiwu@hdu.edu.cn (W.W.)

**Keywords:** calorimetric microflow sensor, viscous dissipation, vibration amplitude, measure range, sensitivity

## Abstract

The performance of the calorimetric microflow sensor is closely related to the thermal sensing part design, including structure parameter, heater temperature, and operation environment. In this paper, several measures to enhance the performance of the calorimetric microflow sensor were proposed and further verified by numerical simulations. The results demonstrate that it is more favorable to reduce the negative impact of flow separation as the space between detectors and heater is set to be 1.6 μm so as to improve the accuracy of the sensor. With an appropriate gap, the front arranged obstacle of the upstream detector can effectively widen the measure range of the sensor, benefiting from the decrease in upstream viscous dissipation. Compared to a cantilever structure, the resonances can be effectively suppressed when the heater and detectors are designed as bridge structures. In particular, the maximum amplitude of the bridge structure is only 0.022 μm at 70 sccm, which is 53% lower than that of the cantilever structure. The optimized sensor widens the range by 14.3% and significantly increases the sensitivity at high flow rates. Moreover, the feasibility of the improved measures is also illustrated via the consistency of the trend between the simulation results and experimental ones.

## 1. Introduction

Over the past few decades, microflow meters were used in a wide range of applications such as environmental monitoring, indoor air conditioning systems, and biomedical instrumentation [1,2,3]. With the development of MEMS technology since the 1980s [4], thermal microflow meters based on the theory of thermal convection [5] have acquired high attention because of their advantages such as high sensitivity, high accuracy, and high reliability [6]. In general, thermal microflow sensors consist of two components [7], namely, the heater and the detector. The heater is designed to provide heat while the detector measures the change in heat caused by the fluid flow and, thus, reflects the magnitude of the flow [8]. Given the different operation modes of the heater, thermal microflow sensors can be divided into three categories: hot-wire/hot-film type, calorimetric type, and time-of-flight type [9,10,11]. The last type is more expensive for measuring flow and the hot-wire/hot-film type does not perform well in applications targeting the measurement of low-flow fluids [12]. Therefore, calorimetric thermal microflow sensors hold a greater market value.

An initial study in this field was conducted by O. Tabata [13] who proposed a silicon flow sensor with an on-chip temperature sensing element. To decrease the influence of gas flow on sensor performance, L. Dong et al. [14] proposed methods of arranging obstacles with different sizes and shapes around the sensor. In a later work, Lee D. et al. [15] designed a calorimetric microflow sensor for measuring drug delivery flow rates, with the advantage of detectable fluid flow rates as low as 0.1 mL/h. In another study, Fang Z. et al. [16] innovatively proposed to replace the conventional thermistor with the change of electrochemical impedance in an ion solution, and thereby, the thermal microflow sensor devised on this basis can reach the flow detection level of μm/s. For the first time, Xie Y. et al. [17] investigated the effect of the various structural parameters on the performance of the MEMS calorimetric wind sensor by combining both simulation and experiment. Furthermore, two packaging designs were systematically researched for a calorimetric microflow sensor by Xu et al. [18], open-space and probe-with-channel types, and the results revealed that the latter type can achieve greater sensitivity. With continuous efforts made by predecessors, impressive progress has been realized in the field of the calorimetric microflow sensor [19]. However, there are still three shortcomings of the microchannel calorimetric microflow sensor studied here [20]: (1) the measure range is not wide enough due to the tendency of the calorimetric microflow sensor to be prone to saturation at high flow rates [21,22]; (2) the sensitivity needs to be improved especially in the case of large flow rates [23]; (3) the structural vibrations result in worse robustness under high flow rates [24].

To address the above-mentioned issues, in our group, some corresponding measures to improve the sensor performance were presented by means of numerical simulation. First of all, the space between the detectors and the heater was adjusted, and then, the optimal space was selected based on the range and sensitivity; moreover, the cause of the sensor failure was explained from the perspective of flow separation. Secondly, an obstacle was adopted at the front of the upstream detector and the optimal gap between them was determined, which was also discussed thoroughly from the viewpoint of viscous dissipation. Lastly, the suppression of resonance using different support forms of the thermal sensing part was evaluated to upgrade the robustness of the sensor. Additionally, the performance of the initial model and the improved model were compared, which could indicate the microflow sensor with wider measure range, higher sensitivity, and better robustness.

## 2. Method

### 2.1. Geometric Model

The 3D model of the thermal microflow sensor is shown in Figure 1; the computational domain of the sensor consists of a fluid domain (microchannel) and a solid domain (thermal sensing part), of which the microchannel consists of an MEMS wafer with a small cavity (L × W × H = 400 μm × 400 μm × 28.2 μm) and a CMOS wafer with a large cavity (L × W × H = 800 μm × 400 μm × 52 μm) eutectic bonded together. The thermal sensing part is composed of three thermistors, including one heater (L × W × H = 62.4 μm × 350 μm × 6.2 μm) and two detectors (L × W × H = 54.2 μm × 350 μm × 6.2 μm). Every thermistor contains an identical serpentine structure with a wire width of 6.4 μm and 3.98 μm for the heater and detectors, respectively, and the wire space of each thermistor is 1.6 μm. The types and thicknesses of the materials applied in the heater and detectors are 5 μm SOI layer, 0.2 μm molybdenum layer, and 1 μm aluminum nitride layer. Moreover, the structural diagram of the thermal sensing part with the obstacle and different forms of support is given in Figure 2.

The thermal microflow sensor model investigated in this paper is at the micron level, and therefore, its continuity at the microscopic scale needs to be considered by the Knudsen number in Equation (1):(1)$$Kn=\frac{\lambda }{D_{h}}$$
where λ denotes the mean free range, and $D_{h}$ denotes the characteristic length. From the model dimensions above, Kn is much less than 0.01; hence, the medium can be regarded as continuous.

Subsequently, three conservation laws need to be followed to build a numerical model. The equation for the conservation of mass is as shown in Equation (2):(2)$$\frac{\partial \rho }{\partial t}+\nabla \cdot \left(\rho \vec{U}\right)=S_{m}$$
where ρ denotes the fluid density, $\vec{U}$ denotes the vector flow rate, and $S_{m}$ denotes the mass source term and is set to be zero in this work, since no phase transition problem is involved.

The equation for the conservation of momentum is as shown in Equation (3):(3)$$\frac{\partial \left(\rho \vec{U}\right)}{\partial t}+\nabla \cdot \left(\rho \vec{UU}\right)=-\nabla \vec{p}+\nabla \cdot \vec{\tau }+\rho \vec{g}+\vec{F}$$
where $\nabla \vec{p}$ denotes pressure gradient, $\vec{g}$ denotes gravitational acceleration, $\vec{F}$ denotes volume force, and $\vec{\tau }$ denotes stress tensor.

The equations for the conservation of energy are described in Equations (4) and (5) [18]:(4)$$\frac{\partial }{\partial t}\left(\rho E\right)+\frac{\partial }{\partial x_{i}}\left(u_{i}\left(\rho E+p\right)\right)=\frac{\partial }{\partial x_{i}}\left(k\frac{\partial T}{\partial x_{i}}-\sum_{j'}h_{j'}J_{j'}+u_{i}\left(\tau_{ij}\right)\right)+S_{h}$$
(5)$$E=h-\frac{p}{\rho }+\frac{u_{i}^{2}}{2}$$
where k is the heat transfer coefficient, h is the specific enthalpy, T is the temperature, and $J_{j'}$ is the diffusion flux of component j. The first three terms on the right-hand side of the above equation represent the energy transport due to heat transfer, component diffusion, and viscous dissipation, respectively, and $S_{h}$ is the volumetric heat source term. The numerical scheme and discretization method are the same as in our previous work [25]. All the calculations were run in Fluent 12.0 software.

### 2.2. Operating Principle

The operating principle of the sensor can be briefly described as follows. In the absence of gas, the temperature field generated by the heater is symmetrically distributed. In the presence of gas, the temperature field will deviate, and the resistance values of the detectors will change accordingly, which is captured by the Wheatstone bridge built on the MEMS wafer and then transmitted to the interface circuit on the CMOS wafer.

In addition, the following points need to be made concerning the thermal behavior of the material. Molybdenum belongs to the PTC thermistor with a TCR coefficient of 2.22 × 10^−3^ (22–90 °C). The molybdenum layer of the heater generates heat under the control of a constant temperature circuit. The molybdenum layer in the upstream and downstream detectors can convert temperature signals into electrical signals. Moreover, the heat transfer mode mainly involves heat conduction, heat convection, and heat radiation. The focus of this paper is on the conjugate heat transfer between fluid and solid, so the heat transfer mode of heat convection is mainly considered.

### 2.3. Initial Model and Validation

The size of the computational grid used in this case refers to the study in the literature [26]. For the 160 sccm flow rate, calculations show that the temperature difference between their upstream and downstream detectors does not exceed 0.1% for various grid numbers 700,000, 1,000,000, 2,000,000. Therefore, the final 700,000 mesh model is adopted. The physical model is chosen as laminar attributing to the Reynolds number of 771.98 at 165 sccm and the material parameters of the computational domain are shown in Table 1. Since the Mach number is larger than 0.3, this is a compressible flow. The boundary conditions can be assumed:

Inlet boundary conditions: velocity-inlet, initial temperature 295.15 K.

Outlet boundary conditions: pressure outlet, temperature 295.15 K, zero gauge pressure.

The temperature of the outer wall and the heater molybdenum layer are set to be 295.15 K and 351.15 K, respectively.

Furthermore, the simulation results of the computational domain model are compared with the experimental results of the microflow sensor, as shown in Table 2. It can be seen that the relative error between the experimental results and the simulation results fluctuates around 7.5 (K − V) K^−1^ under different flow rates, indicating that the model is highly reliable and that subsequent simulation processing can be carried out based on this model. The relative error is defined as ${\mathrm{TD}}_{\mathrm{Num}}-{\mathrm{TD}}_{\mathrm{Exp}}/TD_{Exp}$. It should be noted that the experimental results should be transfered to temperature difference (TD) according to the calibration curve in ref. [18].

## 3. Results and Discussion

In this part, several measures are simulated to further improve the sensor performance based on the initial model. Moreover, the effectiveness of the improvement measures will be discussed based on sensitivity, range, and vibration amplitude. Here, the sensitivity is defined as the slope of the linear fit between the two data points, which is expressed in Equation (6):(6)$$S_{T}=\frac{\Delta T}{\Delta Q}$$
where ΔT and ΔQ denote the temperature difference and flow difference between two adjacent data points, respectively.

### 3.1. The Effect of Heater–Detector Spaces on Sensor Performance

Considering the wire space of each thermistor, the minimum space between the heater and detectors is set to be 1.6 μm according to the manufacturing process. The sensitivity response to flow rate for Space = 1.6 μm, 2.6 μm, 3.6 μm, 4.6 μm, 5.6 μm, and 6.6 μm, respectively, is shown in Figure 3. Obviously, the sensitivity of the sensor at different spaces tends to decrease as the flow rate increases within 40–70 sccm. And the smaller the space, the higher the sensitivity that can be achieved at the same flow rate. It can be seen that the sensitivity of the sensor at Space = 1.6 μm for the same flow rate is higher than the other groups. In addition, the measure range of the initial model (Space = 4.6 μm) is 65 sccm by numerical simulations. Note that the measurement range is defined as the maximum flow rate when the temperature between the upstream and downstream detectors increases with the flow rate [18,20]. The measure range decreases when Space > 4.6 μm, and the measure range can reach 70 sccm when Space = 1.6 μm and 2.6 μm. Accordingly, Space = 1.6 μm is the most reasonable owing to the high sensitivity and large measure range of the sensor.

The vector for 80 sccm is presented in Figure 4, which visualizes the internal flow of the upstream detector front area at high flow rates. The middle part of the computational domain formed by the small MEMS cavity and the large CMOS cavity divides the flow into two regions, the upper part of which is smaller in volume and slower in flow rate. Additionally, there is a significant flow separation phenomenon caused by an adverse pressure gradient [27], which will result in two consequences: on the one hand, the heat in the upstream detector region at high flow rates cannot be carried to the downstream in time, and on the other hand, viscous friction will be generated between the return flow fluid and the upstream detector. These two factors will contribute to the temperature difference between upstream and downstream decreasing gradually with the increase in flow rate, and consequently, the measure range of the sensor will narrow.

This section investigates the sensor performance in the condition of different gaps between the obstacle and the upstream detector. The influence of obstacles with different gaps on the upstream detector temperature under different flow rates is shown in Figure 5. It is worth noting that in all three cases the upstream detector temperature will rise, which will cause the temperature difference between the upstream and downstream detectors to drop, leading to sensor failure. However, the corresponding flow value at the saddle point increases to 80 sccm in the presence of the obstacle while the upstream detector temperature starts to rise after reaching the saddle point (65 sccm) in the absence of the obstacle. Therefore, the addition of the obstacle can effectively widen the measure range of the sensor. The viscous dissipation on the top surface (y = −1 μm) of the detectors and the heater in the microchannel for these three cases is given in Figure 6. The viscous dissipation value in the case without the obstacle exhibits an approximate step change at the front end of the upstream detector (x = −90 μm), indicating that the upstream detector is subjected to a large viscous shear. It can also be found that the viscous dissipation at the front end of the upstream detector is lower at Gap = 2 μm, which is what allows the sensor to achieve a larger measure range compared to without the obstacle. And the viscous dissipation maximum is lowest at Gap = 2 μm, which is the reason why the sensor sensitivity at Gap = 2 μm in Figure 5 is slightly greater than at Gap = 5 μm.

### 3.2. The Effect of Obstacles on Sensor Performance

The distributions of fluid velocity near the thermal sensing part under different gaps between the obstacle and the upstream detector for the flow rate of 80 sccm are shown in Figure 7. The fluid flow process can be described as follows: first, the fluid flows in from the inlet; then, the fluid passes through the sudden expansion place and flows around the thermal sensing part; finally, the fluid diffuses to the outlet. When an obstacle is added at the forward end of the upstream detector, the fluid will first flow around the obstacle and then through the upstream detector. It can be seen in Figure 7a that the fluid flows to the upstream detector with a higher velocity after flowing from the sudden expansion place. However, the fluid velocity near the upstream detector decreases significantly with the addition of obstacles and the fluid velocity near the thermal sensing part decreases as the gap between the obstacle and the upstream detector diminishes. In conclusion, the added obstacle is able to keep the viscous dissipative region away from the upstream detector, and hence, the effect on the temperature difference is reduced. The height and width of the obstacle will influence the temperature of the upstream detector. But the dominant factor is the addition of the obstacle.

### 3.3. The Effect of Support Form on Sensor Performance

The support form of the thermal sensing part is composed of a cantilever or a bridge, and the effect on the robustness of the sensor is studied. Ansys Workbench was employed for the calculation of the natural frequency of the structures. Firstly, we calculated the first six natural frequencies of structures with different support forms (Figure 8). For the cantilever structure, the values are 0.016546 MHz, 0.016564 MHz, 0.025263 MHz, 0.025267 MHz, 0.029456 MHz, and 0.029471 MHz; for the bridge structure, the values are 0.021342 MHz, 0.021353 MHz, 0.032665 MHz, 0.03267 MHz, 0.03657 MHz, and 0.036589 MHz. Then, the corresponding excitation frequency bands, which are 0.015–0.03 MHz for the cantilevered structure and 0.02–0.037 MHz for the bridged one, were selected based on the calculated natural frequencies. Finally, the vibration frequency and amplitude response to the cantilevered and bridged structures at 40 sccm and 70 sccm are illustrated in Figure 9. The vibration amplitude of the two structures appears as a step response when the excitation frequency is close to the natural frequency of the structure; it is, therefore, crucial to consider the stability of both support forms in the case of resonance. Comparing the vibration amplitude magnitude of the same structure at different flow rates, particularly when resonance is generated, it can be noted that the vibration amplitude of the structure increases with increasing flow rate so that the suppression of vibration in both structures at high flow rates needs to be further regarded. The amplitude of the bridged structure is always lower than that of the cantilevered one at 70 sccm, which implies that the bridged structure can inhibit the vibration effectively and its robustness is better than that of the cantilevered support form.

### 3.4. Improved Model Simulation Results and Validation

The improved model can be established after the parameters of the improvement measures are determined based on the initial model. The space between detectors and heater is 1.6 μm, the gap between the added obstacle and the upstream detector is 2 μm, and both the heater and detectors are a bridge structure. The temperature difference–flow response curves of the two models are shown in Figure 9a. It can be seen that the measure range of the improved model is effectively widened. Meanwhile, the sensitivity of the initial model and the improved model is also displayed in Figure 9b, where it is clear that the sensitivity of the improved model is always better than the initial model at high flow rates. In addition, the experimental data of the initial model agree well with the simulation data of the improved model, indicating both the feasibility of the improvement measures and the reliability of the improved model.

## 4. Conclusions

Based on our previous work, we propose several efficient methods to enhance the performance of calorimetric microfluidic sensors. Numerical results and comparison with experimental results indicate that

(1)The performance of the microflow sensor proposed in this present work is obviously better than the original one.(2)The preferential space between the detectors and heater is 1.6 μm due to the better sensitivity and greater measuring range.(3)An arranged obstacle at the front of the upstream detector can efficiently improve the performance of the calorimetric microflow sensor because the viscous dissipative region can be kept away from the upstream detector. A gap of 2–5 mm between the obstacle and the upstream detector can satisfy the engineering requirements.(4)The bridge structure of the heater and detectors could ensure better robustness of the sensor due to the decreased vibration amplitude.

## Figures and Tables

**Figure 1 sensors-23-07413-f001:**
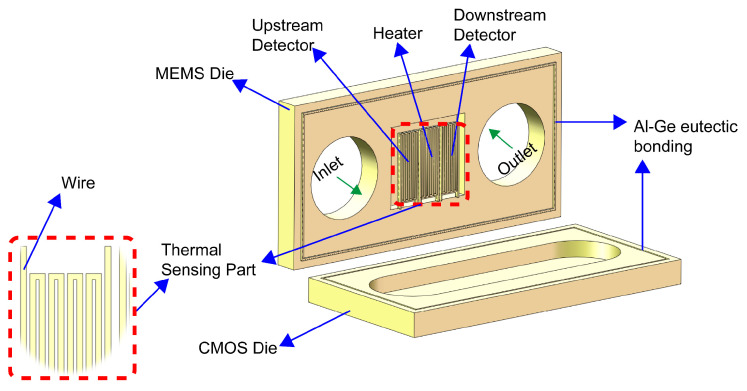
The 3D model of the thermal micro flow sensor.

**Figure 2 sensors-23-07413-f002:**
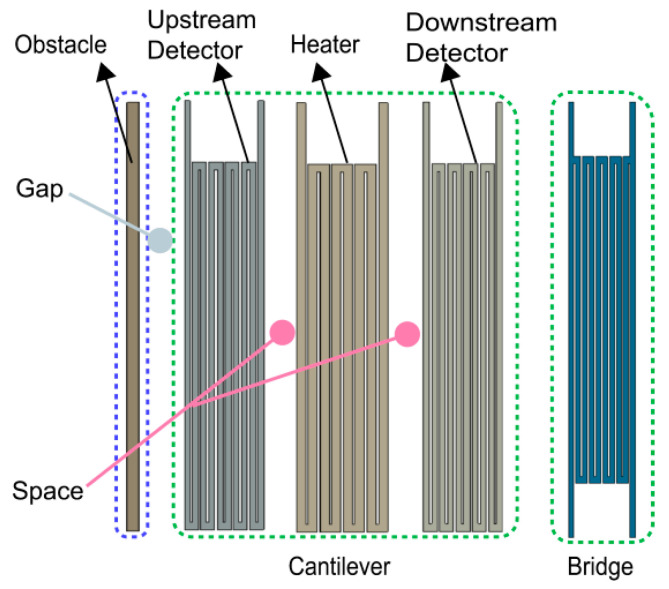
The structure of solid with the obstacle and different support forms as well as various gaps and spaces.

**Figure 3 sensors-23-07413-f003:**
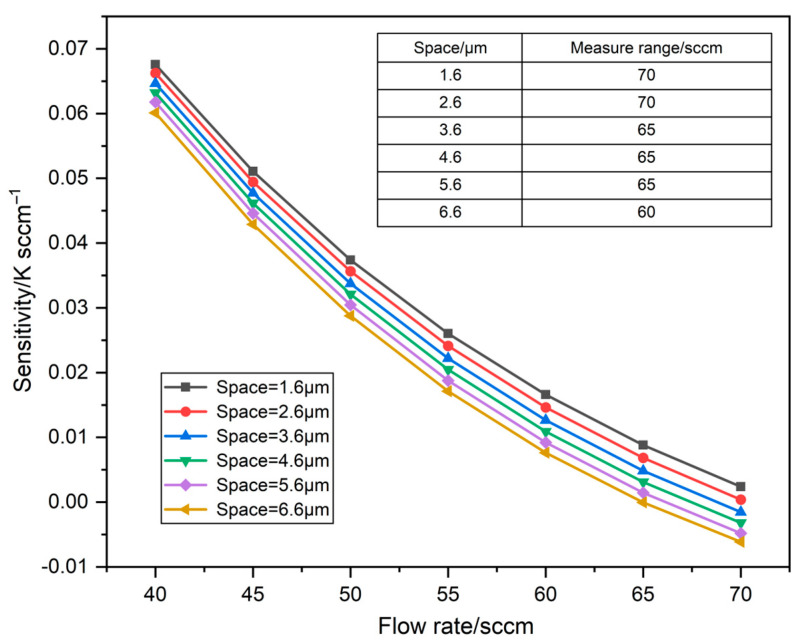
The sensitivity to flow rate at various spaces.

**Figure 4 sensors-23-07413-f004:**
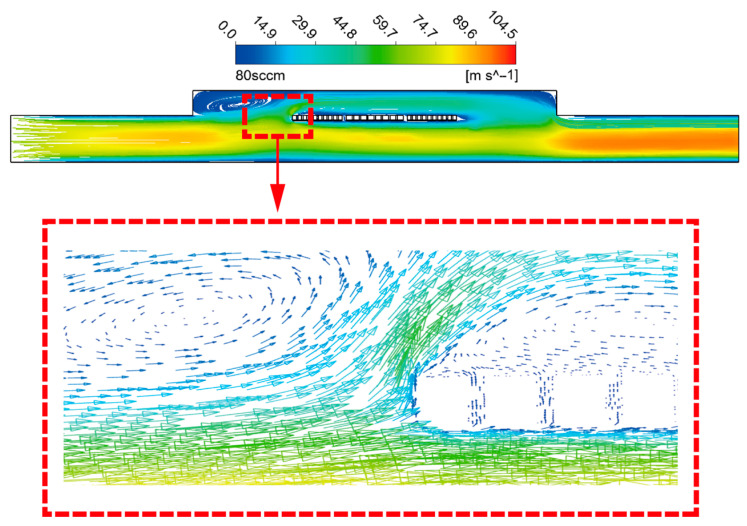
The vector of the upstream detector front area at 80 sccm.

**Figure 5 sensors-23-07413-f005:**
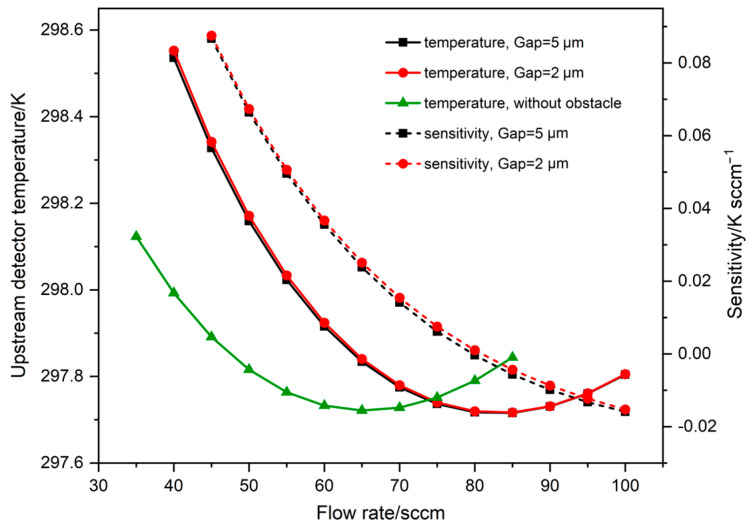
The effect of different flow rates on the upstream detector temperature and sensitivity of the sensor with different gaps between the obstacle and the upstream detector.

**Figure 6 sensors-23-07413-f006:**
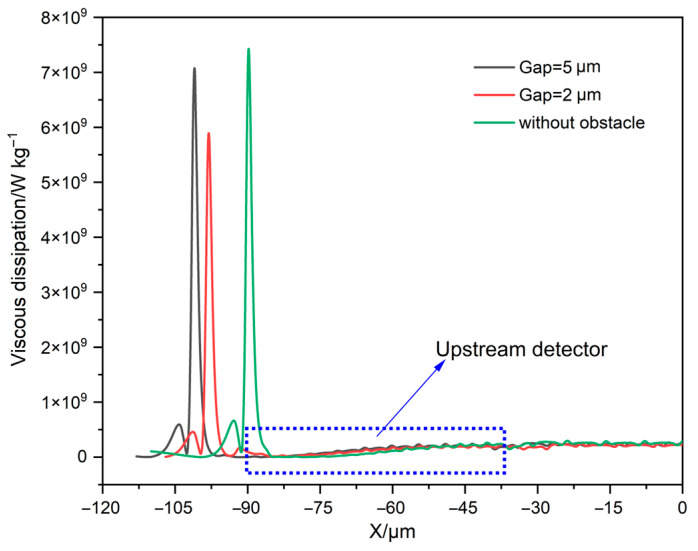
The viscous dissipation on the top surface (y = −1 μm) of detectors and heater at 80 sccm for different gaps.

**Figure 7 sensors-23-07413-f007:**
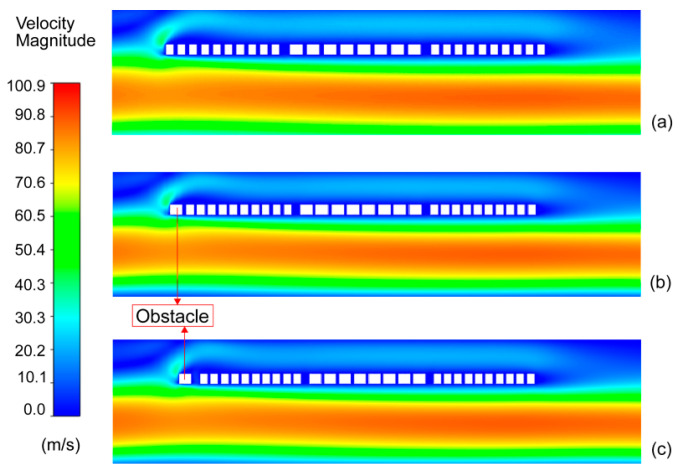
The contours of fluid velocity under different gaps. (**a**) Without obstacle; the gap between the obstacle and the upstream detector is (**b**) 2 μm or (**c**) 5 μm.

**Figure 8 sensors-23-07413-f008:**
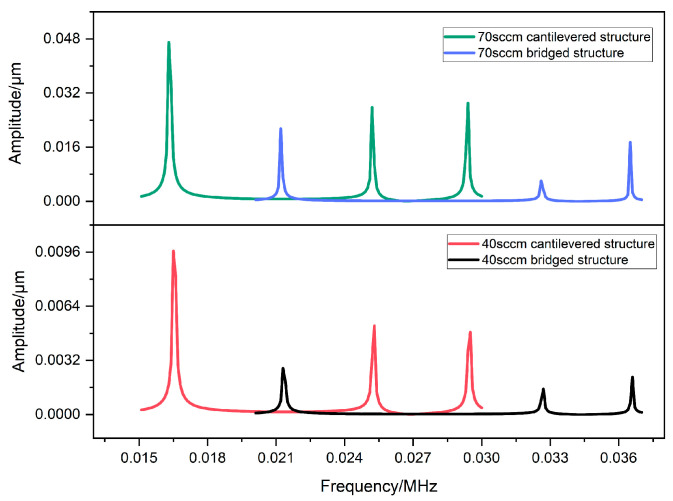
The vibration frequency and amplitude response to cantilevered and bridged support forms at different flow rates.

**Figure 9 sensors-23-07413-f009:**
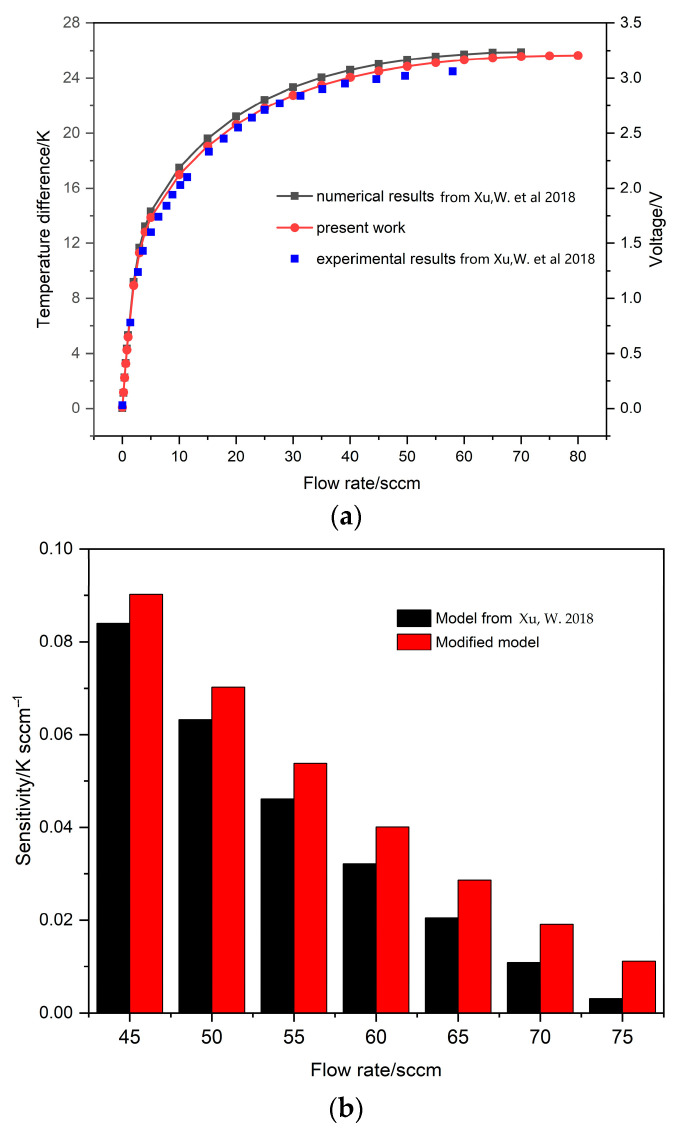
The temperature difference and voltage response to flow rates (**a**); the sensitivity response to flow rates (**b**) [20].

**Table 1 sensors-23-07413-t001:** Material parameter properties in the computational domain.

Domain	Density kg m^−3^	Specific Heat kJ kg^−1^ K^−1^	TC Wm^−1^ K^−1^	Viscosity N m^−2^s
N_2_	Ideal gas law	1032.8	0.0242	Sutherland’ law
AlN	3260	30.1	285	-
Mo	10,280	250	138	-
Si	2329	702	124	-

**Table 2 sensors-23-07413-t002:** Comparison between the initial model simulation results and experimental results [23].

Flowrates/sccm	5	10	15	20	25	40	50
Num results/K	14.31	17.49	19.61	21.21	22.41	24.59	25.33
Exp results/V	1.6	2.03	2.33	2.55	2.71	2.95	3.02
Relative Error	7.9	7.6	7.4	7.3	7.3	7.3	7.4

## Data Availability

No applicable.

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
