# Peer review of "Investigation on the Effective Measures for Improving the Performance of Calorimetric Microflow Sensor"

_sensors, 2023, doi:10.3390/s23177413_

Round 1

Reviewer 1 Report

1. The article only provides simulation results for space larger than 1.6 um and lacks simulations for space smaller than 1.6 um. This seems insufficient to support the conclusions of the article.

2. How is the initial simulation space set to 1.6um determined?

3. The measurement range of the sensor is calculated based on what?

4. Should the two paragraphs below Figure 4 be placed in Section 3.2?

5. Does the height and width of the obstacles have any impact on the sensor output?

6. The conclusion regarding the effect of obstacles on sensor performance is based only on the comparison of two sets of data, with gaps of 2 um and 8 um. It is recommended to add more sets of data to enhance the reliability of the conclusion.

Polish the English Expressions in some sentences.

Author Response

Response to the comments of Reviewer 1

  1. The article only provides simulation results for space larger than 1.6 um and lacks simulations for space smaller than 1.6 um. This seems insufficient to support the conclusions of the article.

Response: Thank you for your constructive comments. The minimum space between the upstream and downstream detectors and the heater was set to be 1.6 μm because the wire spaces of thermistor is 1.6 μm according to the manufacture process. Some explanations have been added in the revised manuscript.

  1. How is the initial simulation space set to 1.6um determined?

Response: Thanks. The setting value of 1.6μm refers to the wire space of thermistor.

  1. The measurement range of the sensor is calculated based on what?

Response: Thanks. The determination of the sensor range is related to the principle of the sensor. The range of the sensor is related to the temperature difference between the upstream and downstream detectors. At small flow rates, the temperature difference scales linearly with the flow rate. At moderate flow rates, instead, the temperature difference increases nonlinearly with the flow rate. However, this relation can no longer hold, that is, the temperature difference does not increase with the flow rate anymore beyond a critical flow rate. The measurement range of the sensor is determined as the critical flow rate. Some explanations have been added.

  1. Should the two paragraphs below Figure 4 be placed in Section 3.2?

Response: Thanks for your suggestions. Figure 5 illustrates the internal flow of the upstream detector front at high flow rates when no obstacles are added, which in turn reveals the mechanism of the sensor failure. The effect of obstacles on sensor performance is investigated in Section 3.2. It is therefore reasonable to place Figure 5 in Section 3.1.

  1. Does the height and width of the obstacles have any impact on the sensor output?

Response: Thanks for the constructive comments. This manuscript only investigates the effect of the addition of an obstacle and space to the upstream detector on the sensor correlation performance, excluding the effect of structural parameters. We will pursue a parametric study in the future. Of course, the main effect of adding an obstacle is to reduce the viscous dissipation at the upstream detector front. Some explanations have been added in the revised manuscript.

  1. The conclusion regarding the effect of obstacles on sensor performance is based only on the comparison of two sets of data, with gaps of 2 um and 8 um. It is recommended to add more sets of data to enhance the reliability of the conclusion.

Response: Thank you very much for the recommendation. The main effect of the obstacle is to reduce viscous dissipation at the upstream detector front. When the space between obstacle and the upstream detector is changed, it does not significantly improve the performance of the sensor any more because the heat generation due to the viscous dissipation will not affect the temperature of upstream detector anymore (see Fig. 6). Therefore, it can be concluded that the gap between the obstacle and upstream detector has no significant effect on the output of the sensor.  

Reviewer 2 Report

Comments to the Authors

 Title:  Investigation on the effective measures for improving the  performance of calorimetric microflow sensor.

 The authors investigate numerically the performance of the calorimetric microflow sensor, including preferential space between heater and detectors and obstacles arranged at a suitable position at the front of the upstream detector, as well as the bridge structure was adopted for thermal sensing part, by an improved  dynamic mathematical model.  The innovation of the present work is presented in a well-documented article. The presentation of proposed numerical simulation and the validation of results must be improved. The results and conclusions must be improved. Also, there are some points that need further improvement to make the paper suitable for publication.

Following are the remarks. In view of the remarks, I recommend for major revision of the manuscript.          

 Page 4 – section: 2.3 Initial model and validation

The authors mentioned: “…..The size of the computational grid used in this case refers to the study in literature [25] and the final mesh comprises two million cells, as shown in Fig. 3. The physical model is chosen as laminar attributing to the Reynolds number of 771.98 at 165 sccm and the material parameters of the computational domain are shown in Table 1. Besides, the wall temperature contact with wafers directly was set to be 295.15 K, and the initial temperature of the heat source was then set to be 351.15 K.. .….”

 -       Enrichment of bibliography regarding the governing equations used is suggested.

-       Since the final accuracy of a numerical experiment is a function of the grid quality and time step have the authors conducted a grid and time step independence study and if so, it should be included.

-       It should be given more details about numerical solution of governing equations and the initial - boundary conditions (flow rate, pressure etc..)

The authors mentioned: “…..Furthermore, the simulation results of the computational domain model are compared with the experimental results of the microflow sensor, as shown in Table 2. It can be seen that the relative error between the experimental results and the simulation results fluctuates around 7.5 (K-V) K-1 under different flow rates, indicating that the model is highly reliable and that subsequent simulation processing can be carried out based on this model. .….” An additional analysis and presentation are required for this conclusion. The authors should mention the accuracy with which the predicted values of the model matched with those of the other works by being more statistically analysis. Also, the reference to this particular experiment is missing.

 Page 9 – section: 3.4 Improved model simulation results and validation

-        For the better presentation and analysis, the authors should be included more details about the improved model.

- Perhaps the authors should mention more statistical analysis for the accuracy with which the predicted values of the present work  matched with those of the experiment and initial numerical model.

 Page 10 – section: 4 Conclusions

Poorly written conclusions that do not allow the necessity and innovation of the present work to emerge and be adequately presented to stimulate further research.

Author Response

Response to the comments of Reviewer 2

The authors investigate numerically the performance of the calorimetric microflow sensor, including preferential space between heater and detectors and obstacles arranged at a suitable position at the front of the upstream detector, as well as the bridge structure was adopted for thermal sensing part, by an improved dynamic mathematical model. he innovation of the present work is presented in a well-documented article. The presentation of proposed numerical simulation and the validation of results must be improved. The results and conclusions must be improved. Also, there are some points that need further improvement to make the paper suitable for publication.

Following are the remarks. In view of the remarks, I recommend for major revision of the manuscript.

Response: Thanks for the positive comments. We revise in light of constructive comments. The innovation of this work is to propose three efficient measurements to improve the performance of the presented sensor.

Page 4 – section: 2.3 Initial model and validation

The authors mentioned: “…..The size of the computational grid used in this case refers to the study in literature [25] and the final mesh comprises two million cells, as shown in Fig. 3. The physical model is chosen as laminar attributing to the Reynolds number of 771.98 at 165 sccm and the material parameters of the computational domain are shown in Table 1. Besides, the wall temperature contact with wafers directly was set to be 295.15 K, and the initial temperature of the heat source was then set to be 351.15 K.. .….”

 - Enrichment of bibliography regarding the governing equations used is suggested.

Response: Revised.

- Since the final accuracy of a numerical experiment is a function of the grid quality and time step have the authors conducted a grid and time step independence study and if so, it should be included.

Response: Thanks. Grid independence validation related content and convergence conditions have been added.

- It should be given more details about numerical solution of governing equations and the initial - boundary conditions (flow rate, pressure etc..)

Response:Thanks. The details of the numerical solution and the initial boundary conditions of the model are given in the revised manuscript.

The authors mentioned: “…..Furthermore, the simulation results of the computational domain model are compared with the experimental results of the microflow sensor, as shown in Table 2. It can be seen that the relative error between the experimental results and the simulation results fluctuates around 7.5 (K-V) K-1 under different flow rates, indicating that the model is highly reliable and that subsequent simulation processing can be carried out based on this model. .….” An additional analysis and presentation are required for this conclusion. The authors should mention the accuracy with which the predicted values of the model matched with those of the other works by being more statistically analysis. Also, the reference to this particular experiment is missing.

Response: You are right. How does the comparison in Table 2 need to be performed for demonstration?  The experiments related to this sensor have been carried out in the previous work [18, 20], and the relevant data have been presented by way of citation. The aim of the present work is to propose efficient measurements to improve the sensor performance.

Page 9 – section: 3.4 Improved model simulation results and validation

- For the better presentation and analysis, the authors should be included more details about the improved model.

Response: Thanks for the constructive comments. Details of the numerical solution and the initial boundary conditions of the model are added in the revised manuscript.

- Perhaps the authors should mention more statistical analysis for the accuracy with which the predicted values of the present work matched with those of the experiment and initial numerical model.

Response: Your suggestion is of great importance to our work, and some analysis and discussion are added in the revised manuscript. Moreover, we will use neural networks to further analyze and optimize the sensors in the future.

Page 10 – section: 4 Conclusions

-Poorly written conclusions that do not allow the necessity and innovation of the present work to emerge and be adequately presented to stimulate further research.

Response: The conclusions have been rewritten.

Reviewer 3 Report

In this manuscript, the authors present a comprehensive investigation of the factors influencing the performance of the calorimetric microflow sensor, with a focus on the thermal sensing part design. The proposed measures to improve the accuracy and widen the measuring range of the sensor are intriguing and supported by numerical simulations. However, there are some points that need further clarification and expansion:

1.      What kind of numerical tool has been applied to solve the governing equations?

2.      The mesh dependency test has not been found.

3.      In the conservation of mass equation (2), the mass source term Sm should be for the phase change problem. In the current simulation, I don’t see any phase change phenomena.

4.      The boundary conditions (velocity inlet? Pressure outlet? Inflow mass rate?) of this simulation haven’t been found.

5.      All the numerical schemes and discretization methods are missing.

6.      In Table 1, the density of N2 is an ideal gas. However, there is no ideal gas law shown in governing equations. Is it a compressible flow?

7.      Please list the definition of relative error in Table 2.

8.      How to calculate viscous dissipation?

9.      How does the author calculate the natural frequency of the structures?

Author Response

Response to the comments of Reviewer 3

  1. What kind of numerical tool has been applied to solve the governing equations?

Response: Finite volume methods based on the Ansys Fluent platform were used to solve the equations. Some details are added.

  1. The mesh dependency test has not been found.

Response: Thanks for the constructive comments. Related content on grid independence verification has been added in the revised manuscript.

  1. In the conservation of mass equation (2), the mass source term Sm should be for the phase change problem. In the current simulation, I don’t see any phase change phenomena.

Response: This mass source term is set to zero in the present simulations, since no phase transition problem is involved. Some explanations have been added in the revised manuscript.

  1. The boundary conditions (velocity inlet? Pressure outlet? Inflow mass rate?) of this simulation haven’t been found.

Response: Relevant reference to boundary conditions have been added to the revised manuscript.

  1. All the numerical schemes and discretization methods are missing.

Response: Thanks. The numerical scheme and discretization method are the same as in our previous work and will be cited in the revised manuscript.

  1. In Table 1, the density of N2 is an ideal gas. However, there is no ideal gas law shown in governing equations. Is it a compressible flow?

Response: Thanks. It is a compressible flow since the Mach number is greater than 0.3. Some explanations have been added in the revised manuscript. 

  1. Please list the definition of relative error in Table 2.

Response: Revised according to the comment.

  1. How to calculate viscous dissipation?

Response: The viscous dissipation term is included in the 3D model in Fluent.

  1. How does the author calculate the natural frequency of the structures?

Response: Ansys Workbench was employed for the mode analysis.

Round 2

Reviewer 2 Report

The authors have carried out revisions. The manuscript is now acceptable for publication.

Reviewer 3 Report

The author has answered the concerns in the last manuscript.